

# Temperature Trends, Climate Attribution and the Nonstationarity Question

Ross McKitrick[1], Timothy Vogelsang[2], John Christy[3]

[1]Department of Economics, University of Guelph, Guelph ON N1G 2W1 Canada ORCID 0000-0002-2257-0675
[2]Department of Economics, Michigan State University, East Lansing MI United States
[3]Earth System Science Center, University of Alabama in Huntsville, Huntsville AL, USA

*Correspondence to*: Ross McKitrick (ross.mckitrick@uoguelph.ca)

**Abstract.** The standard trend model for measuring climate warming assumes error terms are mean-reverting and stationary. But the climate econometrics literature has argued that if anthropogenic forcing is a dominant driver of climate, temperature

trends must have nonstationary (unit root) error terms, which may be considered a "fingerprint" for anthropogenic forcing. Herein we explain this paradox and apply some tools from time series econometrics to resolve it.  We formalize a previously proposed hypothesis for why past results have been unclear, namely that temperatures contain both a nonstationary forcing component and a stationary "weather noise" component that may bias unit root tests towards over-rejection. Our analysis yields a diagnostic method for assessing whether this problem matters in practice. We apply unit root tests to observed and

modeled temperature series at surface and tropospheric layers. We find observed temperatures are stationary around a trend after allowing for a single structural break in trend, with no evidence of testing bias due to weather noise. Unit root tests applied to model-generated temperatures also indicate trend stationarity however we find evidence of testing bias due to weather noise. This implies that time series models for climate attribution need to deal carefully with the requirements for establishing cointegration. We discuss the implications for understanding the relationship between greenhouse gas forcing

and atmospheric temperatures over time.

**Key words:** Global climate change, unit roots, nonstationarity, attribution, cointegration

**Short Summary:** The climate econometrics field has shown that attribution of warming to anthropogenic forcings requires

temperature data to have a property called "nonstationarity" whereas trend detection assumes the data are stationary. Detailed testing shows temperatures are best described as stationary deviations around a linear trend. This is not consistent with anthropogenic forcings being the dominant driver of observed trends over time in the empirical framework commonly used in climate econometrics.





## 1 Introduction

It is common in climate analysis to estimate trends and confidence intervals using a linear model with a first-order autoregressive (AR(1)) error structure:

$$y(t) = \alpha_0 + \alpha_1 t + e(t),$$
$$e(t) = \rho e(t-1) + \epsilon(t), \qquad\qquad (1)$$


where $y(t)$ is temperature at time $t$, $e(t)$ is the random component around the trend, $\rho$ is the AR1 coefficient, and $\epsilon(t)$ is a zero-mean error term that is typically assumed to be Gaussian. The $\alpha's$ are trend coefficients usually estimated, along with $\rho$, using a maximum likelihood estimation (MLE) method. The random component $e(t)$ is covariance stationary when $|\rho| < 1$. This is the model used, for example, by the Intergovernmental Panel on Climate Change (IPCC) in its Sixth Assessment

Report (Gulev et al. 2021 Table 2.4). Identical calculations are shown in Table 2.4 of the IPCC Fifth Assessment Report, Table 3.2 in the IPCC Fourth Assessment Report etc. Meanwhile, there is a longstanding consensus in the climate econometrics literature that since anthropogenic forcings due to greenhouse gases are nonstationary whereas natural forcings are stationary, global average temperatures must have a random component that is a nonstationary unit root process (stochastic trend) in which case $\rho = 1$ in equation (1) (Cummins et al. 2022, Dergiades et al. 2016, Beenstock et al. 2016,

Kaufmann et al. 2010, Mills 2009). Nonstationary of the error terms in equation (1) is thus a "fingerprint" for the dominance of anthropogenic forcings. If true, then the maximum likelihood estimators of the $\alpha's$ and $\rho$ have nonstandard sampling distributions which invalidate standard MLE inference and the corresponding confidence intervals for estimated trends as used by the IPCC and others. In contrast, if $|\rho| < 1$, then IPCC trend confidence intervals may be valid but the attribution arguments used by, for example, Dergiades et al. (2016) and Cummins et al. (2022) that attempt to establish that

anthropogenic forcings are the primary drivers of global warming are invalid because they use tools from cointegration analysis which require that $e(t)$ be a unit root process ($\rho = 1$).

Putting it more simply, two active strands of the empirical climate literature invoke contradictory assumptions about the nature of temperature data. Methods for detecting significance of trends require temperature data to be covariance-stationary around a linear trend ($|\rho| < 1$) whereas attribution of the trend to anthropogenic forcing requires the data to be nonstationary

($\rho = 1$). Interestingly each side can point to empirical literatures supporting its preferred assumptions about $\rho$. In part because of the barriers of technical language these contradictory strands of literature have developed in isolation from each other for at least the past decade without resolution.

This paper makes two contributions. We first explain why the value of $\rho$ matters and why conflicting results can be found in the literature, which we do by formalizing an intuitive explanation that has previously been proposed, namely that





temperatures are composed of a nonstationary forcing component and stationary "weather noise" which biases the unit root test result toward non-detection of the stochastic trend. We will show that this hypothesis can account for conflicting test results and also implies a testing strategy that can identify whether the bias is likely present. We then apply the testing

strategy to modeled and observed temperature series from the surface and the lower and mid-troposphere (herein LT and MT, respectively). The climate econometrics literature has thus far exclusively focused on the surface record despite it being in some respects unsuitable for identifying the time series properties of climate processes.

Before summarizing our empirical findings we should clarify some terminology. Following standard practice, a random

component with a unit root is labeled an I(1) process which is short-hand for "integrated of order 1". This label follows from the fact that when $\rho = 1$, $e(t)$ becomes covariance stationary upon first-differencing. A covariance stationary random component does not need to be differenced to induce stationarity and is labeled an I(0) process. The term 'trend stationary' refers to a trending series with a stationary (I(0)) random component. There is no established label for a trending series with an I(1) random component, so we adopt the label 'trend nonstationary-random-component' or the more compact label 'trend

nonstationary' for such a process. Some readers may ask why we are not using the label 'stochastic trend' for a trending-I(1) series. The reason is that in the time series literature the label 'stochastic trend' only refers to the random component because a mean zero I(1) component can sometimes generate observed series that 'appear' to be trending up or down. The label 'stochastic trend' says *nothing* about whether or not the series has a deterministic trend component. We will avoid the potentially confusing label of 'stochastic trend' and use the less confusing label 'trend nonstationary'.


In our empirical analysis we find statistical evidence that observed globally-averaged temperature series at the surface are, for the most part, trend stationary especially when we allow the deterministic trend component to have one potential structural break at an unknown date. The evidence in support of trend stationarity is stronger for the troposphere series LT and MT. The tropospheric temperature record provides a useful point of comparison especially given that results are less

clear cut in the surface record. The composite weather noise-plus-climate forcing model that we sketch implies that ensemble averaging and de-noising methods should result in stronger statistical evidence of trend nonstationarity. We clearly see this pattern with model-generated temperature data but, interestingly, the opposite occurs with observational data. We also discuss implications for the analysis of cointegration between temperatures and anthropogenic forcings where, again, conflicting evidence exists in the literature.


Throughout this paper we use language from time series analysis, specifically *nonstationarity*, *trend stationarity*, *cointegration*, *unit roots* and *orders of integration*. For readers who are unfamiliar with these terms Appendix A provides basic definitions. The next section outlines the specific analytical questions to be explored and explains our testing framework. Section 3 presents results of unit root tests and Section 4 presents the cointegration analysis. Section 5 provides

discussion and conclusions.



## 2 Summary of the issues

### 2.1 Temperatures and Nonstationarity

If average observed temperatures $y(t)$ are trend stationary, then estimating deterministic trend coefficients using a model like equation (1) or a variant that allows for more general autocorrelation processes (e.g. McKitrick and Vogelsang 2014) is

straightforward. Autocorrelation-robust testing of hypotheses about the trend slope coefficients is well established (e.g. Vogelsang (1998) and Bunzel and Vogelsang (2005). As indicated by the above references to IPCC reports, trend stationarity is an unstated working assumption for much of applied climatology in the sense that it is routine to report estimated trends and confidence intervals and to draw inferences about trend magnitudes without first testing for nonstationarity in the random component (testing for a unit root). While not the focus of this paper, trend stationarity is also

an unstated working assumption in the detection of changes in extreme events since comparing a recent temperature deviation to historical deviations from trend is only meaningful if the variation around the trend is constant over time. A trend nonstationary series has a random component with a variance that is increasing with time making deviations far from the trend more likely as time evolves.

Because the trend stationary (I(0)) assumption is so ubiquitous and essential in practice, it is remarkable to note, as observed by Dergiades et al. (2016), that "the literature contains considerable evidence that the temperature time series is I(1)." Anthropogenic forcings, herein denoted $F^A(t)$, have additionally been found to be either I(1) or I(2) around the trend but never I(0), whereas natural forcings, herein denoted $F^N(t)$, are typically found to be I(0) around the trend (Kaufmann et al. 2013, Beenstock et al. 2013). Forcings are all measured in a common unit (Watts per square meter) and the standard IPCC

modeling framework assumes that the combined effect is given by the sum of the individual components (Myhre et al. 2013). The "signal detection" framework for making causal attribution connecting climate change to greenhouse gases assumes $y(t)$ can be represented as a linear function of the summed forcings (Cummins et al. 2022), but if the forcings add up to a trend nonstationary series and $y(t)$ is trend stationary the coefficients on the forcings must be zero (see section 2.2 below). Consequently, attribution in a time series context depends on the random component of $y(t)$ sharing the same order

of integration as the random components of the summed forcings, and on the series cointegrating to yield a stationary residual.

A practical challenge for assessing whether a series is trend stationary or trend nonstationary is that there are many unit root tests and results can be sensitive to the selection of the autoregressive lag used by many unit root tests. Recall the AR(1)

model for $e(t)$ given by equation (1) can be written $e(t) = \rho e(t-1) + \epsilon(t)$ where $|\rho| < 1$ is required for $e(t)$ to be stationary. By subtracting $e(t-1)$ from both sides and defining $\pi = \rho - 1$ we see that unit root tests can be built around a regression of the first differences of a series (denoted with $\Delta$) on its own lagged values, for example





$$\Delta e(t) = \pi e(t-1) + \epsilon(t). \tag{2a}$$


Estimating $\pi$ by ordinary least squares (OLS) and testing the null hypothesis $\pi=0$ (equivalently $\rho=1$) with a $t$-statistic yields the so-called Dickey-Fuller test (Dickey and Fuller 1979). This is typically called a test of a unit root. Under the unit root null hypothesis this $t$-statistic has a non-standard distribution so the usual standard normal table cannot be used. Instead, the critical values are taken from tables of the Dickey-Fuller distribution. Because $e(t)$ is unobserved, the trend component must

first be estimated and removed with $e(t)$ in (2a) replaced with residuals $\hat{e}(t)$. Alternatively, an autoregressive model for $y(t)$ can be specified in the same form as (2a) but with an intercept and time trend included:

$$\Delta y(t) = \alpha_0^* + \alpha_1^* t + \pi y(t-1) + \epsilon(t). \tag{2b}$$

If the nonstationary I(1) null is rejected, the alternative implied by either equations (2a) or (2b) is that y(t) is a trend stationary I(0) process. Many variations on this basic test need to be considered in order to arrive at robust conclusions, including the following.

- The error term $\epsilon(t)$ may have additional autocorrelation which can be handled by adding lagged values of

145       $\Delta \hat{e}(t)$ or $\Delta y(t)$, giving the so-called Augmented Dickey-Fuller (ADF) regression and corresponding ADF unit root $t$-statistic. The number of lagged first differences must be chosen by the researcher ideally based on some statistical information about the autocorrelation structure of $\epsilon(t)$. In practice the value of the ADF unit root $t$-statistic can be highly sensitive to lag selection.

- The specification of the deterministic trend function is very important. Suppose $\alpha_1 \neq 0$ but the time trend

150       regressor, $t$, is left out of (1) or (2a). Then it is well known (Perron, 1988) that the ADF unit root test will suffer from systematically low power (biased toward non-rejection when the series is trend stationary).

- This lack of power holds in general when the deterministic component is under specified. Suppose that $y(t)$ is trend stationary but the trend undergoes a structural break (slope shift and/or level shift) at a known or unknown date. Failure to allow for this will cripple power and bias the test towards over-reporting unit roots.

155       When allowing for a structural break in the trend, the researcher must either impose the date based on exogenous information or use a data-dependent method for detecting the date of the break. Zivot and Andrews (1992) pointed out that choosing a break date and treating it as known after looking a plots of the data can bias





the test towards over-reporting stationarity (I(0)) and they recommended a procedure for endogenous determination of the break date based on minimizing the value of the ADF *t*-statistic (maximizing the chance of rejecting the unit root null). Vogelsang and Perron (1998) extended this approach and developed an additional method for estimating a break date based on maximizing the value of an *F* test of the no-break null. We will discuss these approaches in Section 3. If allowing a structural break in the trend changes an I(1) result to an I(0) result, then a unit root is rejected in favor of trend stationarity.[1]

- While under-specification of the deterministic trend substantially reduces power of a unit root test, over-specification also reduces power but to a much lesser extent. If a break is allowed in the trend of series when it is not needed, this will use up two degrees of freedom, but the test will still have power to detect stationarity of the random component. Rejection of the null, in such cases, indicates loss of power is not an issue.

- The method used to estimate the deterministic trend component matters for the power of unit root tests. The ADF approach uses OLS estimators of the trend parameters. Under the null hypothesis of a unit root the trend parameters can be estimated more precisely using generalized least squares (GLS). These issues combine: allowing a break point still requires choice of lag length to control for serial correlation. Elliott et al. (1996) proposed a variant of the ADF approach using a GLS step that estimates the trend coefficients separately then carries out the ADF regression using the GLS detrended data. Deterministic regressors are not included in the ADF regression. This method, known as ADF-GLS, can have higher power to detect stationarity when $\rho$ is relatively close to one (but is less than one).

What happens if different unit root tests yield conflicting results? Because results can depend on the choice of lag length, best practice is to use an objective sequential method in which high order statistically insignificant lags are dropped from the ADF approach until a statistically significant coefficient on a lag is found. See Vogelsang and Perron (1998, page 1078) for details. Also the deterministic trend component should have a flexible functional form that is rich enough to encompass the trending features of temperature series. Given that the temperature series we analyze are increasing over time, the base-line

---

[1] The limit to this principle, though, is that as more breaks are permitted and more lags are included, degrees of freedom are used up and a unit root test can lose power, thereby failing to detect stationary random components.





deterministic trend is linear as in equation (1). To improve power we also implement unit root tests allowing a single structural break in the trend at an unknown data. This step is also reasonable on physical grounds with regard to long temperature series in a system subject to changes in internal modes of oscillation that can induce periodic changes in global trends (e.g. Kravtsov et al. 2018). We use data dependent methods to choose the break date and the impact of the data dependent method on the unit root test is reflected in the critical value of the ADF or ADF-GLS $t$-statistics.

The same issues of interpretation discussed hitherto arise with model-simulated temperatures, which we denote $x_i(t)$ for models $i = 1, \ldots, N$. Some of the issues can be clarified using the framework of Cummins et al. (2022, herein C22) which explored the link between cointegration and attribution by considering the following system of equations:

$$y(t) = \Phi(B)F(t) + e(t) \tag{3}$$
$$x_i(t) = \Phi'(B)F(t) + e_i(t) \tag{4}$$
$$x_i^A(t) = \Phi'(B)F^A(t) + e_i^A(t) \tag{5}$$

where $\Phi(B)$ and $\Phi'(B)$ are rational functions of the backshift operator, $F(t)$ is the time series of the sum of all historical forcings (GHG's, solar, aerosols, etc.), $x_i^A(t)$ is the hindcast model temperature series based only on the sum of anthropogenic forcings $F^A(t)$, and the $e$'s are stationary I(0) error terms.[2] The necessary assumptions applying to equations (3—5) are not clearly stated in C22 but are as follows:

> **Assumption A1**: $y(t)$ is I(1);
>
> **Assumption A2**: $x_i(t)$ and $x_i^A(t)$ are both I(1);
>
> **Assumption A3**: The forcings $F(t)$ and $F^A(t)$ are I(1);
>
> **Assumption A4:** observed and model-generated temperatures cointegrate with the associated summed forcings so that the error terms in equations (3)—(5) are all I(0).

C22 present a lemma which is an adaptation of the Beveridge-Nelson decomposition to establish a theorem which says that if Assumptions A1-A4 hold for equations (3—5), a non-zero vector of cointegrating coefficients can be found that yields a stationary linear combination, $r(t)$, of the $y$'s and $x$'s, and the coefficients from a regression of $y(t)$ on $x_i(t)$ and $x_i^A(t)$ are therefore consistent estimators of signal coefficients $\beta$ which, in turn, can be used to reveal the magnitude of the effects of the forcings on the observed climate. Details on this regression are given in Section 3.5.

---

[2] C22 allow $p$ separate forcings but the reduced version shown here does not lose generality.





Rather than proving the general consistency of the signal detection regression framework, the C22 theorem makes clear how strong the underlying assumptions need to be for attribution regressions to be meaningful. It follows from their theorem that if Assumption A1 does not hold, namely if $y(t)$ is I(0) but the other assumptions hold, then it must be the case that $\Phi(B)$ in equation (3) equals zero and since that implies the $\beta$'s are also zero, $y(t)$ cannot be a function of the forcings. Their theorem also assumes the error terms in equations (3)—(5) are I(0) which in turn requires cointegration between all temperatures (observed and modeled) and the forcings. Having imposed that assumption, the signal detection regression can be interpreted as using the model temperatures as linear proxies of the forcings which were already assumed to drive them and $y(t)$ jointly. The causality interpretation rules out *a priori* the possibility of other variables driving $y(t)$ or of $y(t)$ being an explanatory variable for $x_i(t)$ which could arise through climate model tuning.

C22 make the stronger claim that the existence of a cointegrating vector among $y(t)$, $x_i(t)$ and $x_i^A(t)$ is both necessary and sufficient for the system of equations (3—5) to yield consistent signal detection coefficients. Sufficiency indeed follows from their theorem but not necessity. $r(t)$ could be I(0) if all of $y(t)$, $x_i(t)$ and $x_i^A(t)$ are I(0), in which case equations (3—5) are misspecified. Alternatively if $y(t)$ is I(0) and the simple sum of the forcings $F(t)$ is I(1) but the signals $x_i(t)$ and $x_i^A(t)$ cointegrate with each other, then a cointegrating vector can be found even though $\Phi(B) = 0$ and $y(t)$ is not a function of $F(t)$, thus making any non-zero finding of signal detection spurious. We will encounter this case below.

It is interesting to note that C22 do not provide empirical evidence for whether Assumption A1 holds in their observational sample. Given the importance of this assumption to their cointegration analysis, it is an unfortunate omission. This is particularly true given our empirical results in Section 3 indicating that the observed temperature series used by C22 are trend stationary (I(0) around trend) invalidating Assumption A1.

## 2.2 Unit Roots and Weather Noise in Temperature Data

While temperature data since the late 1800s can appear to be trend nonstationary it does not make sense, in principle, to assume that the climate system itself is driven by random walks (unit root processes), because this would imply that contemporary weather conditions reflect the influence of, say, El Niño events during the last interglacial era as much as ones that occurred recently. It is intuitively appealing, therefore, to suppose that the climate is composed of a stationary component with an additive I(1) forced component. As the latter increases its signal magnitude under rising greenhouse gases, it would then come to dominate later portions of long temperature series. This hypothesis was proposed on an intuitive basis by Kaufmann et al. (2013), who argued that the presence of I(0) weather noise overlaid on an I(1) signal will bias unit root tests to over-reject the I(1) null, which they supported with simulation evidence.



Dergiades et al. (2016) provided empirical evidence for this hypothesis by applying unit root tests to a moving window
along a 600-year paleoclimate reconstruction, showing that the series shifts from I(0) to I(1) in the late 1800s. Unfortunately
these results are weakened by their use of the Wahl and Amman (2007) paleoclimate reconstruction which is a replication of
the Mann et al. (1998) paleoclimate reconstruction, which is heavily dependent on a small set of bristlecone pine tree ring
records from the Great Basin region of the US, with the weighting in the reconstruction artificially inflated by an error in the
method of calculating principal components (North et al. 2006 pp. 106-107, McIntyre and McKitrick 2005). The bristlecone
series in question is not recommended for use in temperature reconstructions because their ring widths are particularly
sensitive to rising atmospheric carbon dioxide levels, implying they are mainly a proxy for forcings rather than temperature
(North et al. 2006, p. 50, Graybill and Idso 1993). The transition from I(0) to I(1) behaviour in the Wahl and Amman (2007)
chart may thus be an artifact of the climate proxy choice.

We can formalize the additive component hypothesis in the following stylized time series model. Suppose a temperature
series is denoted $z_t$ and is the sum of three components: a deterministic trend $\mu_t$, a unit root process $\tau_t$ and an independently
and identically distributed (iid) random error $\omega_t$:

$$z_t = \mu_t + \tau_t + \omega_t \tag{6}$$

where $\omega_t \sim iid(0, \sigma_\omega^2)$, $\tau_t = \tau_{t-1} + v_t$ and $v_t \sim iid(0, \sigma_v^2)$. In this example $\mu_t$ may stand for any low-frequency change that
imparts a trend over the sample period, $\tau_t$ is the influence of anthropogenic forcing (which is I(1)) which is assumed to be
global in scale and common to all temperature series, and $\omega_t$ is trendless weather noise. $\sigma_v^2$ then measures the step sizes of
the unit root (forcing) component and $\sigma_\omega^2$ measures the size of the shocks associated with the weather noise component.


In Appendix B we derive a method for detecting if rejection of an I(1) null is due to noise-induced bias. If a de-noising
treatment can be applied to an I(0) series that reduces the weather noise component, such as ensemble averaging, and if as a
result the unit root test on the averaged series moves towards the I(1) region compared to the average of the individual series
test scores, this provides evidence that the data series may be of the type described by equation (6) and has an underlying
unit root component. If no such change occurs, or if the unit root test score on the averaged and de-noised series moves even
farther into the rejection region, there is no reason to suppose the I(0) result is due to test bias. Ensemble averaging makes
sense as a de-noising method in the modeling context because multiple runs contain weather-related processes such as
simulated El Niño cycles that differ in their timing but not magnitude and therefore cancel out under averaging. By contrast,
even when there are multiple observational series, while they will have idiosyncratic measurement errors due to different
construction methods that may average out, they all share the same timing of major climatic system oscillation events and
these will not average out. Therefore, in addition to averaging across data products we will look at the effect of filtering out
the weather noise attributable to the El Niño Southern Oscillation and Pacific Decadal Oscillation indexes.





Use of data from the LT and MT troposphere layers provides a further check on the results from the surface. Because our
data set is confined to the post-1958 period, during which anthropogenic forcings underwent their modern upward trends,
results using the LT and MT series should reveal the relationship to the forcings even more clearly than the longer surface
record. Also, the climate econometrics literature has focused almost exclusively on global surface temperature records, but
trends in these data are dominated by the land record, which in many regions reflects significant influences of urbanization
and land use change in addition to greenhouse forcing (e.g. Quereda et al. 2016, Ren and Zhou 2014, McKitrick and Tole
2012). These influences attenuate with altitude so if the tendency to finding I(1) components in surface temperatures
genuinely represents greenhouse forcing, the tendency should be stronger in the LT and MT layers. Finally, as noted by
Bruns et al. (2020), it can be difficult to distinguish low sensitivity to GHG's and minimal ocean heat storage versus high
sensitivity and moderate ocean heat uptake. But since, unlike at the surface, temperatures in the troposphere adjust very
rapidly to changes in greenhouse forcing (IPCC AR4 pp. 764-765), mixing of surface ocean heat into deep layers is not a
confounding influence.

## 3    Data and Methods

### 3.1 Data Sets

**Surface temperatures**

We now turn to empirical analysis of some important data sets to illustrate the above issues. The first is taken from C22 and
is denoted herein as the Surface data. It consists of five global annually-averaged surface temperature anomaly series
covering the period 1880-2014 along with climate model-generated analogues from 13 models in the CMIP6 archive. From
each model is obtained an ensemble mean showing the simulated hindcast under all observed forcings. Also available is
another simulation under anthropogenic-only forcings which we will use in the cointegration analysis later. The number of
ensembles per model range from 1 to 65. Table 1 reports the OLS decadal trends in C/decade from regressions of the
observed and all-forcings historical simulation temperature series on an intercept and linear time trend. Figure 1 shows the
time series of the averaged observed and model-simulated series. The observational data products are listed under their
familiar names, but see C22 for the specific sources.



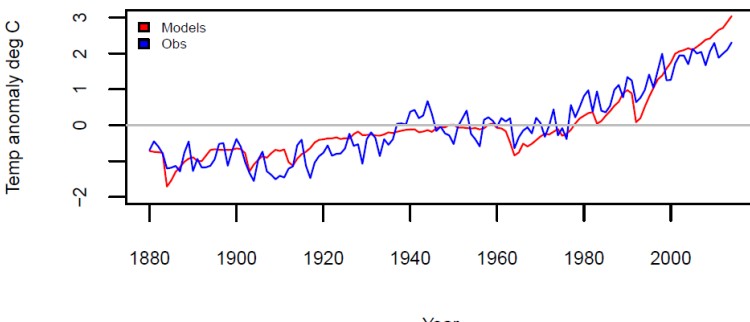

**Figure 1. The averages of five observed (blue) and 13 modeled (red) temperature series 1880-2014.**

|  | Surface Trend |
|---|---|
| Berkeley | 0.080 |
| Cowtan | 0.067 |
| Gistemp | 0.068 |
| Hadcrut | 0.074 |
| NOAA | 0.069 |
| ACCESS.ESM1.5 | 0.046 |
| BCC.CSM2.MR | 0.066 |
| CanESM5 | 0.097 |
| CESM2 | 0.061 |
| CNRM.CM6.1 | 0.054 |
| FGOALS.g3 | 0.082 |
| GFDL.ESM4 | 0.047 |
| GISS.E2.1.G | 0.053 |
| HadGEM3.GC31.LL | 0.050 |
| IPSL.CM6A.LR | 0.084 |
| MIROC6 | 0.047 |
| MRI.ESM2.0 | 0.053 |
| NorESM2.LM | 0.026 |

**Table 1. Top 5 rows: observational data series. Remaining rows: model-generated series. Trend shown is in °C/decade. For details on sources see C22.**





**Tropospheric temperatures**

The second data set is referred to as the Tropospheric data set and was constructed as follows. We use data spanning 1959 to
2021. Observations of layer temperatures are derived from balloon-borne radiosonde records, in which data collected at
specific levels, generally up to 30 hPa (~23 km altitude), are proportionally combined to generate broad, layer-average
Lower-Tropospheric (LT) and Mid-Tropospheric (MT) temperatures corresponding to those that have been monitored by
satellite microwave receivers since late 1978. LT (MT) represents a weighted average from the surface to ~10 km (~18 km)
altitude, often referred to as the temperature of the bulk lower (mid-) troposphere. We use five data products for the
observational record. Two are homogenized data series from the University of Wien (Vienna), the RAdiosonde Observation
Correction using Reanalyses (RAOBCORE v1.7) and Radiosonde Innovation Composite Homogenization (RICH v1.7,
Haimberger et al. 2012). The third is from the U.S. National Oceanic and Atmospheric Administration, the Radiosonde
Atmospheric Temperature Products for Assessing Climate (RATPAC-A v2, Durre and Yin, 2011). The fourth is produced
by the University of New South Wales (UNSW, Sherwood and Nishant 2015).  The fifth is the JRA-55 data set from
Kobayashi et al. (2015). It provides globally-gridded, observationally-constrained pressure-level temperatures from which
the layer-temperatures utilized here may be calculated. The observations employed include radiosonde, satellite soundings,
aircraft, ship and traditional surface temperature measurements which are synthesized into a gridded product through a data-
assimilation process (Kobayashi et al. 2015).

All tropospheric data sets begin in 1959 and go to 2021. We use annual averages as this is the finest time resolution available
from RATPAC for the pressure-level quantities we require. The UNSW series is unavailable after 2016 so that series was
regressed on the other four observational series and the predicted values were taken for the 2017—2021 period.

While polar-orbiting satellites systematically sample the globe twice per day, radiosondes are released only where stations
exist and thus are not evenly distributed.  However, for analyses such as ours (annual averages) the spatial coherence of the
temperature field is strong, mostly due to the ubiquitous tropospheric winds that continually horizontally mix the air.  This
produces, in the annual average, large spatial scales of anomalies.  Thus, relatively few radiosondes are needed to describe
the temperature anomaly on an annual basis which is useful for sparsely covered regions such as the southern oceans
(Hurrell et al. 2000).  Correlation values of annual anomalies of global TMT between radiosondes and satellites exceed
+0.95 (Christy et al. 2018).

We also obtained model runs from 39 CMIP6 models which archived simulation outputs covering the entire time span from
the Lawrence Livermore National Laboratory archive https://pcmdi.llnl.gov/CMIP6/. Models were run using historically-
observed forcings up to 2015 and using RCP4.5 from 2015 to 2021. Table 2 lists the 42 model runs and the 4 observational





series, showing for each the decadal trend in C/decade. In cases in which the model yielded more than one run we used the first run submitted. The model runs are sorted in descending order by LT trend. Figure 2 summarizes the general Tropospheric data patterns. The panels show global MT and LT temperature showing in each case the model ensemble means (red line) and the mean of the observational series (blue line) over the 1959-2021 interval.


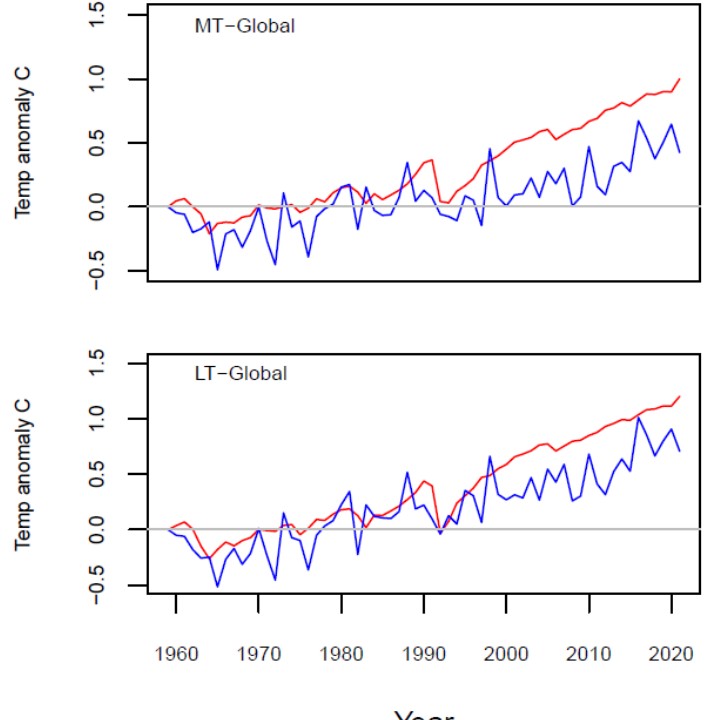

**Figure 2. Averages of five observed (blue) and 39 modeled (red) tropospheric temperature series from 1959 to 2021. All series positioned to start at zero in 1959.**


| Model Full Name | Run Name | LT Global Trend | MT Global Trend |
|---|---|---|---|
| RAOBCORE17 | | 0.163 | 0.107 |
| RICH17 | | 0.186 | 0.133 |
| RATPAC | | 0.191 | 0.118 |
| UNSW | | 0.187 | 0.114 |
| JRA55 | | 0.179 | 0.117 |





| Can5OE | CanESM5-CanOE | r1i1p2f1_gn | 0.352 | 0.303 |
|---|---|---|---|---|
| Can5 | CanESM5 | r1i1p1f1_gn | 0.334 | 0.295 |
| UK10LL | UKESM1-0-LL | r1i1p1f2_gn | 0.281 | 0.193 |
| HadGEM | HadGEM3-GC31-LL | r1i1p1f3_gn | 0.273 | 0.220 |
| KACE | KACE-1-0-G | r1i1p1f1_gr | 0.259 | 0.207 |
| KIOST | KIOST-ESM | r1i1p1f1_gr1 | 0.257 | 0.210 |
| EC_E3V | EC-Earth3-Veg | r1i1p1f1_gr | 0.251 | 0.198 |
| ACCESS_E | ACCESS-ESM1-5 | r1i1p1f1_gn | 0.247 | 0.189 |
| CE2_WAC | CESM2-WACCM | r1i1p1f1_gn | 0.246 | 0.201 |
| MCM_UA | MCM-UA-1-0 | r1i1p1f2_gn | 0.236 | 0.200 |
| GFDL-CM4 | GFDL-CM4 | r1i1p1f1_gr1 | 0.235 | 0.193 |
| FIO | FIO-ESM-2-0 | r1i1p1f1_gn | 0.234 | 0.199 |
| AWI | AWI-CM-1-1-MR | r1i1p1f1_gn | 0.226 | 0.176 |
| CMCC | CMCC-CM2-SR5 | r1i1p1f1_gn | 0.221 | 0.177 |
| NESM | NESM3 | r1i1p1f1_gn | 0.220 | 0.171 |
| CE2r3 | CESM2 | r3i1p1f1_gn | 0.217 | 0.174 |
| FGOALS_f3 | FGOALS-f3-L | r1i1p1f1_gr | 0.217 | 0.176 |
| CNRM_E2 | CNRM-ESM2-1 | r5i1p1f2_gr | 0.214 | 0.148 |
| IPSL6A | IPSL-CM6A-LR | r1i1p1f1_gr | 0.214 | 0.182 |
| ACCESS | ACCESS-CM2 | r1i1p1f1_gn | 0.213 | 0.172 |
| NOR_LM | NorESM2-LM | r1i1p1f1_gn | 0.212 | 0.166 |
| CIESMa | CIESMa | r1i1p1f1_gr | 0.211 | 0.175 |
| FGOALS_g3 | FGOALS-g3 | r1i1p1f1_gn | 0.209 | 0.167 |
| NOR_MM | NorESM2-MM | r1i1p1f1_gn | 0.203 | 0.161 |
| GFDL-ESM4 | GFDL-ESM4 | r1i1p1f1_gr1 | 0.201 | 0.157 |
| MPI_L | MPI-ESM1-2-LR | r1i1p1f1_gn | 0.196 | 0.151 |
| MRI_E2 | MRI-ESM2-0 | r1i1p1f1_gn | 0.193 | 0.155 |
| CNRM_C61r5 | CNRM-CM6-1 | r5i1p1f2_gr | 0.188 | 0.132 |
| BCC | BCC-CSM2-MR | r1i1p1f1_gn | 0.184 | 0.133 |
| EC_E3 | EC-Earth3 | r24i1p1f1_gr | 0.178 | 0.139 |
| MIROC_2L | MIROC-ES2L | r1i1p1f2_gn | 0.177 | 0.137 |
| IITM | IITM | r1i1p1f1_gn | 0.177 | 0.148 |
| MPI_H | MPI-ESM1-2-HR | r1i1p1f1_gn | 0.172 | 0.128 |
| GISSE21G | GISS-E2-1-G | r1i1p3f1_gn | 0.169 | 0.130 |
| CNRM_HR | CNRM_CM6-1-HR | r1i1p1f2_gr | 0.169 | 0.125 |
| CAMS | CAMS-CSM1-0 | r1i1p1f1_gn | 0.163 | 0.137 |
| INM48 | INM-CM4-8 | r1i1p1f1_gr1 | 0.162 | 0.128 |
| INM50 | INM-CM5-0 | r1i1p1f1_gr1 | 0.155 | 0.111 |
| MIROC | MIROC6 | r1i1p1f1_gn | 0.151 | 0.115 |

**Table 2. Top 5 rows: Observational Tropospheric series trends. Remaining rows: model-generated runs ranked by global LT temperature trend (K/decade).**





The 39 climate model simulations utilized here are from among those accepted for analysis in CMIP6 for which the models are executed in standardized simulations using prescribed forcings so they may be intercompared properly. The LT and MT series for models and observations alike are formed by averaging 13 layer-specific series from the surface to 20 hPa. When considering annual anomalies of global tropospheric temperature, the horizontal sampling by radiosondes is sufficient to reproduce global anomalies with greater than 90 percent variance explained. To generate the brightness temperature as

observed by satellite from a radiosonde, the vertical profile of radiosonde temperatures at standard pressure levels is convolved with a function, each level of which has an appropriate weighting to generate a satellite-like temperature. The use of standard pressure levels as the input from radiosondes has been demonstrated to produce highly consistent annual anomalies of temperature as would be seen from a satellite (Spencer and Christy, 1992).


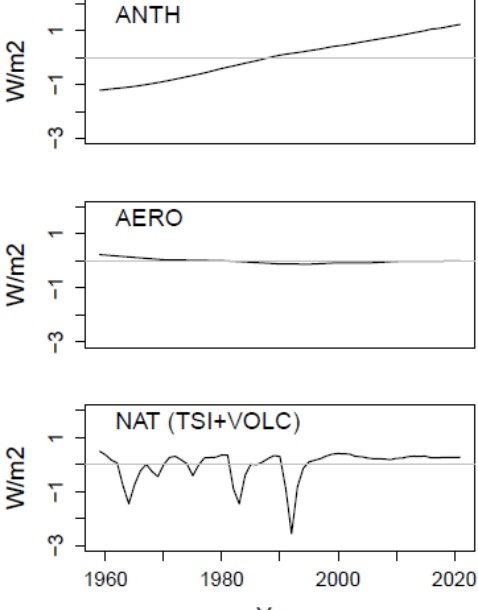

**Figure 3. Forcing series (Watts per square meter) centered on zero means 1959 to 2021. TSI stands for total solar irradiance, VOLC for volcanic aerosols.**

**Forcings**

The CMIP6 process prescribes inputs such as atmospheric carbon dioxide and aerosol levels, solar irradiance etc., then models generate forcing series internally. Consequently there isn't a single prescribed forcing series comparable to those generated for CMIP5 as listed in IPCC Fifth Assessment Report (AR5, 2013) Annex II Table 1.2. We used the latter for this





study, taking them to be representative forcing series that have historically been considered canonical, recognizing however
that internally-generated forcings in CMIP6 models may differ. We confine attention to the post-1958 forcing series in order
to avoid the criticism in Pretis and Hendry (2013) from combining proxy-based and observed historical forcing series
constructed with changing methodologies. The AR5 series report effective radiative forcing anomalies relative to 1750 in
watts per square meter up to 2011, and we also employ extensions to 2017 by Lewis and Curry (2018) (see also their
discussion of minor adjustments to some pre-2011 IPCC estimates). Extensions covering the 2018-2021 interval were done
by using the information in IPCC (2021) Chapter 7 which provides best estimates for forcing changes over 2011-2021. The
change in forcing for the "other greenhouse gas" category was taken to be the sum of methane and nitrous oxide. Forcing
series were converted to "anomalies" or deviations from the mean by centering on zero. The forcings were grouped into
ANTH (the sum of carbon dioxide, other greenhouse gases, tropospheric ozone and land use change) AERO (total aerosols)
and NAT (solar plus volcanic). Figure 3 shows the forcing series.

**Climatic Oscillations**

We used the NINO3.4 index to measure the El Niño Southern Oscillation (ESOI) with the data retrieved from the website of
the National Oceanic and Atmospheric Administration (https://psl.noaa.gov/data/correlation/nina34.data) and, for pre-1948
records,          from          Brian          McNoldy's          website          at          the          University          of          Miami
(https://bmcnoldy.rsmas.miami.edu/tropics/oni/ONI_NINO34_1854-2022.txt). We obtained historical reconstructions of the
Pacific Decadal Oscillation (PDO) index from the website of the US National Oceanic and Atmospheric Administration
(https://www.ncei.noaa.gov/pub/data/cmb/ersst/v5/index/ersst.v5.pdo.dat).

**3.2 Testing Methods**

A simple and intuitive way to assess the time series properties of a variable is to estimate its AR(1) coefficient, $\rho$, after
controlling for trending behavior. If $\rho$ is close to 1, I(1) behaviour is possible. Formally testing that a time series is I(1)
cannot be based on the simple test that the AR(1) parameter is 1 for two reasons. First, if the serial correlation is more
complicated than AR(1), then that correlation needs to be accounted for. Second, as noted previously, the distribution theory
for $t$-statistics that test the I(1) null hypothesis are nonstandard and depend on the specification of the underlying
deterministic trend function.

We employ unit root tests recommended by Vogelsang and Perron (1998) which allow for a single break at an unknown
point with the date of the break determined endogenously by the data. Possible break dates are selected in three ways. The
"maxF-ADF" criterion chooses a break date that maximizes the $F$ test value for testing that the intercept and slope change
parameters are jointly zero in a deterministic trend regression. Using that break date, the unit root hypothesis is tested using





the augmented ADF test. The "minADF" criterion chooses the break date that minimizes the unit root $t$-statistic which is the same as choosing the break date to maximize the chances of rejecting the unit root null. The "minADF-GLS" criterion is the same as minADF except that it based on the ADF-GLS unit root test procedure. In all three cases critical values of the unit root $t$-statistics depend on the method used to choose the break date so that the significance levels of the tests are correct.

Both the ADF-GLS and ADF unit root tests require the choice of a lag length that accounts for additional serial correlation in the fluctuations around the trend. We follow Vogelsang and Perron (1998) and use a general to specific lag approach whereby we begin with five lag terms and test the last included lag coefficient for statistical significance using a standard two-tail t-test at the 10% significance level. The number of lags is reduced one at a time until a statistically significant lag is found or the AR(1) specification is obtained.


For the maxF-ADF test the 10% and 5% left-tail critical values are -4.31 and -4.61 respectively. For the minADF test the 10% and 5% left-tail critical values are -5.08 and -4.82 respectively. For the minADF-GLS test the 10% and 5% left-tail critical values are -3.91 and -3.62 respectively. Because all the unit root tests are left-tail tests, large (in magnitude) negative t-statistics indicate evidence of I(0) fluctuations around the trend against the null of I(1) fluctuations around the trend.

Tests of cointegration are done using the classic Engle-Granger method (Wooldridge 2020), which relies on an ADF test of the residuals from a cointegrating regression.

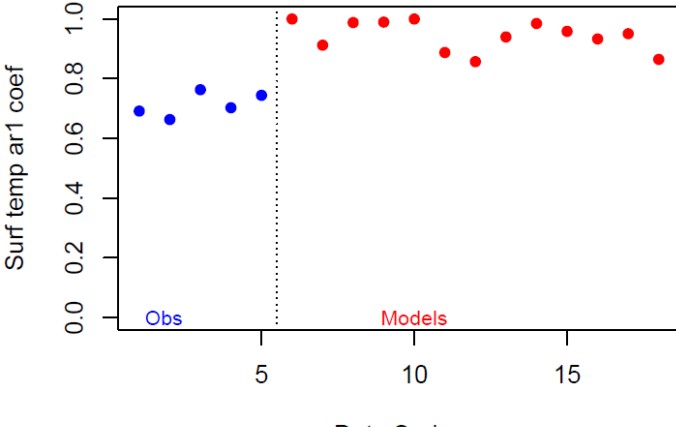

**Figure 4: AR(1) Coefficients for surface data series**





### 3.3 AR(1) Coefficients

Figure 4 shows the AR(1) coefficients (after removing a linear trend) for the Surface data set and Figure 5 shows the same

for the Tropospheric data set. Observed data are in blue and models are in red. There is a clear pattern in which observations

have lower AR(1) coefficients than do models, which implies models exhibit greater persistence than the observed climate.

The AR(1) coefficients for forcings are in Figure 6, and reveals a similar contrast with anthropogenic forcings exhibiting

very high values compared to natural forcings.

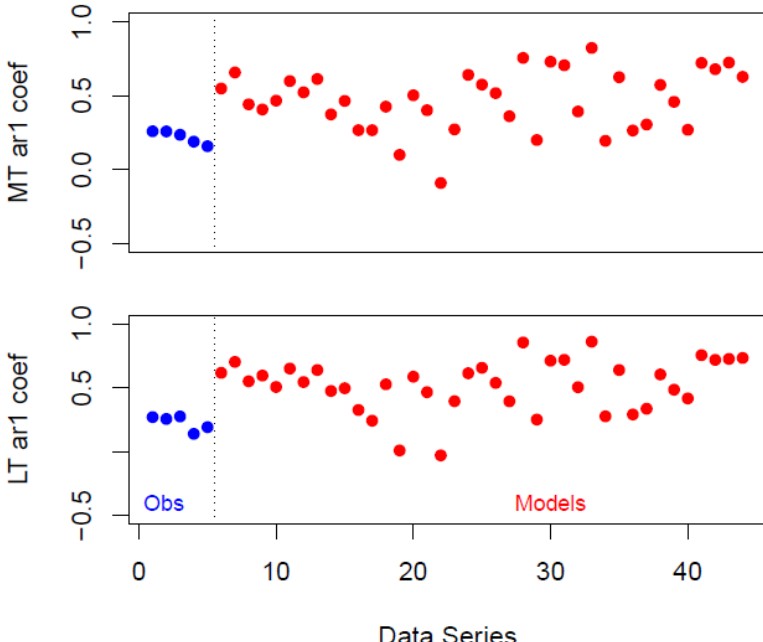

**Figure 5: AR(1) Coefficients for tropospheric data series**





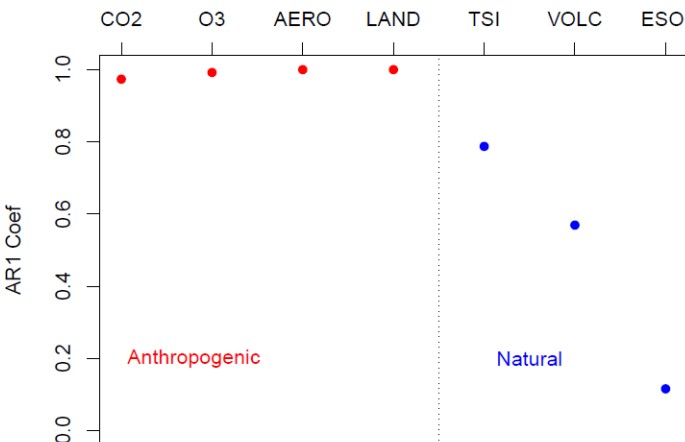

**Figure 6: AR(1) Coefficients for forcings**


### 3.4 Unit root tests

Figures 4 and 5 reveal slight differences in autocorrelation properties between models and observations, and more pronounced differences between anthropogenic and natural forcings. We now turn to series-specific unit root testing and we

note that tests we employ are robust to the form of serial correlation so the results we report herein do not depend on the difference in AR(1) values. In Section 3.5 we will combine modelled and observed series in a cointegrating regression, at which point differences in persistence properties between the two data types need to be formally considered.

|  | Levels | 1st diffs | 2nd diffs |
|---|---|---|---|
| CO2 | -2.828 | -2.363 | -4.915** |
| AERO | -4.048 | -3.241 | -7.230** |
| ANTH | -2.762 | -2.295 | -4.672** |
| NAT | -5.570** | -7.155** | -6.150** |
| 5% cv | -4.610 | -3.610 | -3.610 |
| Coint: ANTH/AERO | No | Yes | |

**Table 3: Unit root tests (maxF-ADF) for forcings and their first and second difference series. The levels case has a linear trend**
**with one structural break. The 1st and 2nd differences cases have a constant with one structural break (level shift). Last row: p-value of test of no cointegration of ANTH (CO2+ozone+land use) and AERO forcings.  ** denotes significant at 5%.**

*Forcings*





Table 3 presents unit root tests for forcings in levels, first differences and second differences. We report the maxF-ADF test, which permits a trend break at an unknown point although the results are not affected by choice of unit root test. The individual anthropogenic forcings are I(2) and the natural forcings are I(0).

We can also ask if the anthropogenic forcings cointegrate down to the I(1) level. The bottom row of Table 3 reports the *p*-
value of a test of whether ANTH and AERO cointegrate. The null hypothesis is no cointegration. In the levels case the *p*-value exceeds 0.1, indicating no cointegration. In the first differences case the *p*-value is below 0.01 indicating cointegration, in other words the forcings combine to an I(1) level, but not by summation since the OLS slope coefficient between them is very small (0.021). Neither the sum ANTH+AERO nor the sum of their first differences is I(0) which means the total anthropogenic forcing is I(2).


*Surface and Tropospheric Temperatures*

"De-noising" of observed data herein refers to filtering the series by averaging over model ensembles or observational data products (which for simplicity we call ensemble averages in both cases) and, in the case of observations, using the residuals from a regression of temperatures on the ESOI and PDO indexes. We employ three unit root tests denoted *maxF-ADF*,
*minADF* and *minADF-GLS*, in each case allowing objective, data-dependent lag and break date identification. The Supplement contains figures showing, for each series, six test scores: the three unit root tests, with and without ESOI/PDO filtering of observations. Also shown in the tables is the result of testing the ensemble average series and the implied integration order *d* for each series.

Figure 7 summarizes the results of the testing procedures by comparing the averages of the series-specific test scores with the tests of the ensemble averages. The top panel shows the MT results, the middle panel shows the LT results, and the bottom panel shows the Surface data results. The maxF-ADF scores are the leftmost column, min-ADF are in the middle and min-ADF-GLS (denoted min-ADFg in the Figure) are in the rightmost column. Each column consists of a pair of results: observations (left) and models (right). Critical values are shown as the dashed lines. In all cases a result below the dashed
line indicates rejection of the I(1) null, implying trend stationarity. The blue dots show the average of observational series unit root scores and the gray open circle shows the unit root score of the de-noised (filtered and averaged) series. The red dots show the average of the model-generated series unit root scores and the orange circle shows the unit root test score of the ensemble mean (note that model-generated data are not filtered to remove ESOI and PDO since those processes are not synchronized across model runs).





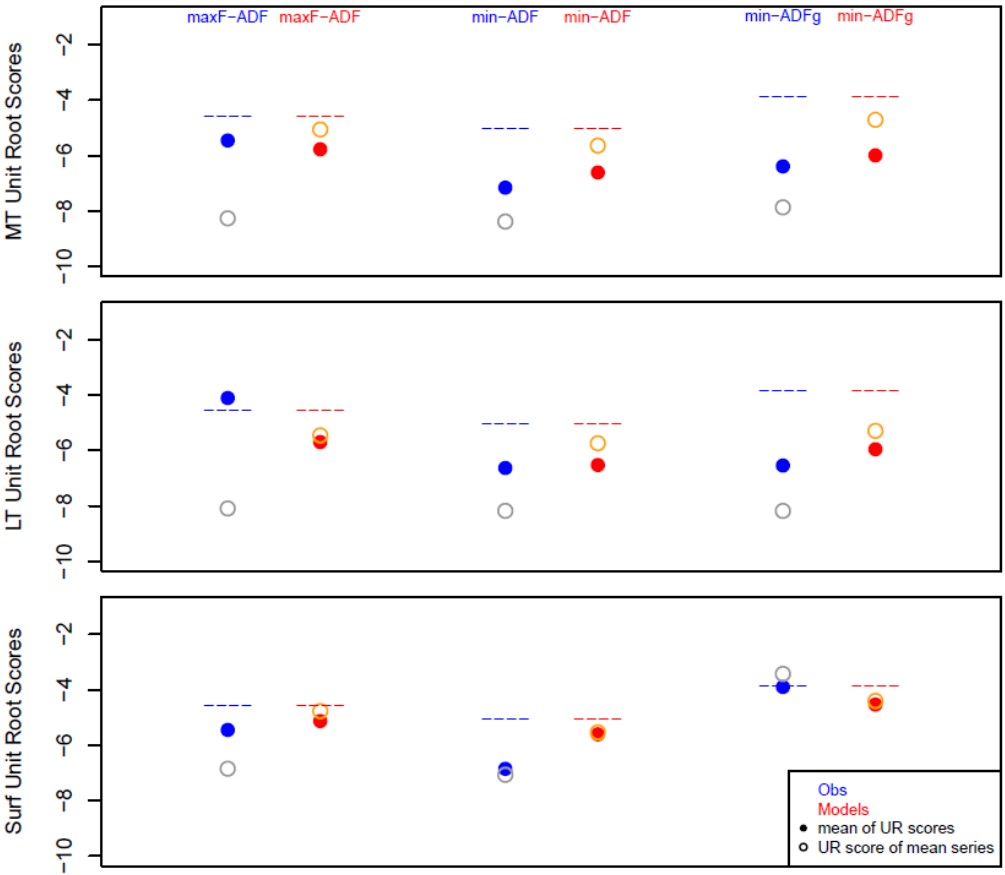

**Figure 7. Summary of unit root test score results. Red: models. Blue: Observed. Horizontal dashed lines show 5% critical value, where a value below the line implies rejection of the unit root null. Open circle: after de-noising.**

Looking first at the observational data, at the surface layer, in two out of three cases the mean test score is below the critical value, indicating the data are I(0). The min-ADF-GLS test places the average right on the rejection threshold. De-noising does not move the test score towards non-rejection in two cases but does by a small amount in the min-ADF-GLS case. Although the break years are not listed, inspection of the test results shows that the maxF-ADF and min-ADF test scores always place the break in the early 1960s, whereas the min-ADF-GLS test places it in the late 1970s or early 1980s. In the LT and MT layers de-noising always moves the test score on observations farther away from the I(1) region. In five of the six cases the average test score is already in the I(0) region. In the MT layer the pattern is clearest: all forms of testing indicate temperatures are I(0) and de-noising moves the test score further into the I(0) region.

The opposite patterns are found using model-generated data. Looking at the MT layer, while the majority of individual series are I(0) trend stationary (see Section 7), ensemble averaging increases the test score and moves it towards the non-rejection



region, consistent with the conjecture in Section 2.3. Indeed the same pattern holds in the LT and at the surface, although in those cases the difference between the average of unit root scores and the unit root score of the average is smaller. The maxF-ADF score places the surface ensemble average at the critical value, but for all other tests the test results are well within the rejection region.

In sum we find that observed temperatures, especially in the tropospheric layers, are trend stationary around a trend with a potential structural break and do not exhibit behaviour suggesting that an underlying additive I(1) process is dominated by an I(0) additive component. Model-generated temperatures, by contrast, also appear to be I(0) but do exhibit behaviour consistent with the additive component hypothesis and testing bias due to weather noise. Anthropogenic forcings, both individually and summed, appear to be I(2), thus differing from observed temperatures by two orders of integration. Using

the composite model (equation 6), this does not rule out the possibility that an anthropogenic forcing-driven unit root exists in the temperature data but it is small enough compared to natural variability and a deterministic low-frequency trend component that it does not drive the outcome of unit root testing, especially in the LT and MT layers.

Results similar to ours were reported in Razzak (2022) who concluded temperatures are trend stationary while anthropogenic

forcings are trend nonstationary, based on application of a large group of unit root tests including those allowing a trend-plus-break, with lag selection determined using information criteria methods. Likewise Storelvmo et al. (2016, Table S1) reported trend stationarity of temperatures and nonstationarity of anthropogenic forcing.

**3.5  Cointegration Analysis**

C22 estimated the following regression:


$$y(t) = \beta_0 + \beta_1 x_1(t) + \beta_2 x_2(t) + \epsilon(t)$$

where $y(t)$ is the HadCRUT5 surface temperature series, $x_1(t)$ is a model-generated temperature series using historical anthropogenic and natural forcings, and $x_2(t)$ is a model-generated temperature series using only greenhouse gas forcing.[3]

Using 13 climate models they tested for cointegration by regressing the first difference of the residuals on the lagged residuals and rejected the null of no cointegration in each case. But C22 did not test whether the forcing signals $x_1(t)$ and $x_2(t)$ are cointegrated with each other. Across all 13 models, using an ADF test we found the residuals of a regression of $x_2(t)$ on $x_1(t)$ are I(0), implying cointegration. But $F(t)$ in equation (3) is, according to IPCC modeling practice, the

---

[3] This assignment of variable names is based on the C22 code. C22 say in the text of their paper that $\beta_1$ measures non-GHG forcings (the "OAN" series in the Jones et al. 2016 notation) which implies a different interpretation of the estimated coefficients. This conflicting notation does not affect the computations discussed here.





unweighted sum of forcings and in each case it is I(1). Consequently at least one I(0) linear combination of $y(t), x_1(t)$ and $x_2(t)$ exists but Assumptions A1 and A4 do not hold and the regression equation (3) is spurious.

Cointegration of $x_1(t)$ and $x_2(t)$ but not with temperatures matches findings in Phillips et al. (2020). They used as explanatory variables the log of the carbon dioxide concentration, which is a measure of anthropogenic forcing, and instrumental readings of total downward infrared radiation at the Earth's surface, which measures all forcings together. They found temperatures were not cointegrated with the log of carbon dioxide or with downwelling radiation, but all three exhibited cointegration, which could arise because the two forcings were cointegrated with each other, a test which unfortunately they did not report.

Balcombe et al. (2019) applied classical and Bayesian structural time series analysis and found mixed evidence for cointegration between forcings and temperature. In a model that did not allow for a trending alternative they could not reject a null of cointegration. But other specifications, including one with a trending alternative and models with lags selected based on information criteria, strongly rejected cointegration and yielded an insignificant coefficient between forcings and temperature. They concluded "previous findings of cointegration between forcing measures and temperatures should be treated tentatively," and proposed that while standard theories of human influence on the climate can be considered directionally valid, time series analysis raises new questions about the actual strength of the connection, a finding confirmed by our analysis.

## 4    Discussion and Conclusion

Two inconsistent views of temperature data coexist in the climate literature. The most familiar one, exemplified by routine tabulations in IPCC reports *inter alia*, is that temperatures are trend stationary and can be described using conventional deterministic trend regressions. An implication of this view however is that signal detection regressions using I(1) forcings are spurious. The other is that temperatures are I(1) which implies methods often used in the climate literature to estimate and generate confidence intervals for trend slopes are invalid, but signal detection regressions are potentially valid if cointegration is found between temperatures and forcings.

Unit root tests results can be found to validate either view, so care must be taken in formulating and applying tests to get robust and objective conclusions. We find that both surface and tropospheric temperature averages appear to be I(0) around a linear trend-with-break model. We applied a battery of unit root tests which treat the break date as endogenous and allow for general serial correlation (beyond the simplistic AR(1) specification) and we find that the predominant inference rejects I(1) behaviour. This is true for both the global surface data and in the LT and MT layers of the troposphere. In the latter case the sample size is smaller, but the rejections are as strong or stronger implying power is not an issue.





We also examined the additive component hypothesis, which posits that temperatures are a composite of stationary and unit root process and the presence of weather noise "tricks" the unit root test into over-rejecting a true null. We formalized the intuition behind this approach and showed that it implies a ranking of results when comparing the average of individual

series' unit root scores versus the unit root score of an averaged and de-noised series. The comparison supports the view that climate models contain both I(0) and I(1) components in model generated temperatures and the unit root test may be biased towards over-rejection of the I(1) null, although even under averaging we do not observe I(1) results. The unit root test results for observed temperatures do not support the additive component hypothesis: an I(1) forcing component, if present, is simply too small to be detected.


If the decision between trend stationarity versus nonstationarity depends on allowing a break in the trend, it is legitimate to ask whether this a justifiable modeling decision. It is important to remember that the estimation method does not impose a break, it only allows one as a possibility and does not impose the timing *a priori*. Since the linear trend model is a restricted version of the trend-plus-break specification it makes more sense to ask if the restriction is justified. There are both statistical

and physical reasons for preferring a model with greater generality. On statistical grounds, the tradeoff is between obtaining increased power to reject a false null and losing power due to the addition of two model parameters. If the break is not needed to describe the trend but the test moves into the rejection region, then there was a net gain in power. On physical grounds it is not reasonable to suppose that the global climate system over a long interval lacks any internal dynamics. Without taking a position on the validity of specific theories regarding such dynamics we merely note that they typically

imply trends which can change in sign and size over multidecadal intervals. Restricting a trend model so it lacks a break term is therefore a less general specification.

We also examined the cointegration approach to attribution. The framework proposed by C22 reveals that strong assumptions are needed for time series signal detection regressions to be valid within a cointegration framework. If

temperatures are I(0) then they cannot be driven by the I(1) components of forcings, including anthropogenic greenhouse gases. Balcombe et al. (2019) and Phillips et al. (2020) found evidence against direct cointegration between anthropogenic forcings and temperatures. The forcings used by C22 are cointegrated with each other but the sum is not cointegrated with temperatures. Neither Balcombe (2022) nor C22 provided test results on whether the observed temperature series used in their analysis are trend nonstationary as required by their assumptions. Our results strongly suggest those series are in fact

trend stationary, implying that the true value of the coefficient on the total forcings is zero.

 Thus in the context of the existing literature we find support for some previous findings and not for others. Dergiades et al. (2016) report that many previous authors have found temperatures are I(1). We would have made similar findings if we used unit root tests that model the deterministic component as either a constant or a linear trend, but when allowing for a structural





break in the trend and controlling for serial correlation, we find temperatures to be trend-plus-break stationary, which suggests the linear trend alternative yields results biased towards the unit root null. We also depart from the climate econometric literature's sole focus on surface temperatures by extending our analysis to include lower- and mid-troposphere layer temperatures. These series are in some respects better suited for the purpose of ascertaining the underlying stationarity properties of the climate. Results for both the LT and MT layers support trend-plus break stationarity (I(0) random

component).

On the one hand, our results imply that conventional trend calculations using temperature data are valid, assuming an adequate correction for serial correlation is applied. On the other hand, our results raise afresh problem that have now been raised by numerous other authors (e.g. Beenstock et al. 2012, Dergiades et al. 2016, Balcombe et al 2019, Razzak 2022) that

if temperatures and greenhouse forcings differ by one or more orders of integration, they are empirically independent and attribution regressions yield potentially spurious causal inferences. The additive component hypothesis points to a potential resolution, namely that an I(1) anthropogenic forcing component is present in temperature data but is too weak to be reliably detected in the context of weather noise and deterministic low-frequency variability.


**Appendix A: Time Series Terminology**

We use the term *stationarity* throughout but we are specifically interested in *covariance stationarity*, which we investigate using the time series concept of integration order. For a given time index $\{t = 1, 2, ...\}$ a variable $x(t)$ is covariance stationary if it has a time constant expected value $E(x)$, a time constant and finite variance $\sigma_x^2$ (or, equivalently, second

moment $E(x^2)$), and covariances between any pair of values $x(t)$ and $x(t - h)$ where $h \geq 1$ depends only on $h$ and not $t$ (Wooldridge 2020). For brevity we drop the "covariance" term in our analysis.

Denote the values $x(1), x(2), x(3), ...$ etc. as "levels" of the series. If $x(t)$ is nonstationary in levels but its first differences, denoted $\Delta x(t) = x(t) - x(t - 1)$, are covariance stationary then $x(t)$ is said to be integrated of order one, or I(1). The "1"

indicates the number of times the series must be differenced to obtain a stationary series. The term "integration" refers to the process by which $x(t)$ can be formed as a cumulative   sum of stationary disturbances: $x(t) = x(t - 1) + \Delta x(t)$. A stationary variable is *integrated of order zero*, i.e. I(0), meaning it needs zero first-differencing to yield a stationary variable. The notation I($d$), where $d$ is a non-negative integer, indicates that a series must be differenced $d$ times to become stationary. The following terms are used interchangeably: I(1), unit root, and stochastic trend. The label "random walk" is used for an

I(1) process in the special case of an AR(1) model.  If $x(t) = x(t - 1) + e(t)$, then the variable $x(t)$ is said to "have", "follow", "contain" or "possess" a unit root, "be" I(1) and "follow" a stochastic trend; these terms all amount to the same





thing. In general the stochastic trend terminology is the least intuitive and is potentially confusing because it says nothing about whether or not the series has a deterministic trend. We avoid this label.

The importance of the distinction between I(0) and I(1) series can be seen by considering the simple first-order autoregressive process $u(t) = \rho u(t-1) + e(t)$ where $e(t)$ is an error term that is uncorrelated over time and has mean zero and constant variance $\sigma_e^2$. If $-1 < \rho < 1$, then $u(t)$ is stationary or I(0) and it is straightforward to show that its variance is $\sigma_e^2/(1-\rho^2)$. After an innovation $e(t)$ at time $t$, $u(t)$ reverts to a mean of zero with the adjustment time controlled by the autoregressive parameter $\rho$. In contrast, if $\rho = 1$, this implies a qualitatively different process: innovations
are permanent, $u(t)$ is no longer mean-reverting and its variance is $t\sigma_e^2$. When $\rho = 1$, $u(t)$ is labeled a *unit root* process and is I(1) because it is a cumulative sum of the stationary time series $e(t)$ via the process $u(t) = u(t-1) + e(t)$.

A series that is I(0) around its trend is called *trend stationary*. There is an important qualitative distinction between a trend series and a unit root series. Even though a trend makes a series nonstationary because its mean depends on time, the change
in the mean across time is predictable and deterministic. Also, departures from the trend are mean-reverting. The change over time in a unit root process is unpredictable and does not have a mean-reverting property. It is also important to keep in mind that a trending series can have I(0) fluctuations around the trend and it can also have I(1) fluctuations around the trend. Departures from the trend for a trend stationary series revert back to trend whereas departures from the trend for a trend nonstationary series do not.


The phenomenon of spurious regression arises when two independent I(1) variables are regressed on one another and the conventional *t*-statistic frequently exceeds 1.96, implying a significant relationship even though the data are unrelated. This well-known fact is easily demonstrated using simulation methods. The underlying problem is that the ordinary least squares estimator of the regression coefficient is inconsistent and the *t*-statistic diverges to infinity (is systematically large).


An exception to the spurious I(1) regression arises under *cointegration*, the case where a group of I(1) variables are related in such a way that a linear combination of them is I(0). This might happen, for example, if the price of a specific commodity is I(1) but is constrained by market forces to remain close to the price of a related commodity which is also I(1). In this case while each price is I(1), the difference in prices is I(0). In other cases there may be a more general linear combination that
yields an I(0) variable. The phenomenon is referred to as *cointegration*. If a linear combination of a group of I(1) variables, i.e. $\lambda_1 x_1(t) + \lambda_2 x_2(t) + \lambda_3 x_3(t)$ etc. can be found which yields an I(0) variable then the *x*'s are said to cointegrate and the $\lambda$'s define a *cointegrating vector*. The implication is that the I(1) variables are related in a long run equilibrium sense: however randomly they move individually over time, they are constrained to return to the equilibrium defined by the cointegrating vector. Authors in the climate econometrics literature have pointed out that cointegration provides a useful
framework for estimating Energy Balance Models (Pretis 2020) and signal detection models (Cummins et al. 2022), but for





this interpretation to be valid, the temperature and forcing series must all be I(1). Alternatively, if temperatures are I(1) and some forcings are I(2) but they cointegrate together to yield an I(1) variable, then the cointegrating framework can still be meaningful.

**Appendix B: Derivation of Test Bias Diagnostic**

Denote the deviations of $z_t$ around its trend as $\tilde{z}_t$, i.e. $\tilde{z}_t = z_t - \mu_t$. From Ng and Perron (2002, herein denoted NP02) we can rewrite $\tilde{z}_t$ as a unit root process with an MA error term: $\tilde{z}_t = \tilde{z}_{t-1} + u_t$ where $u_t = e_t + \theta e_{t-1}$ where $e_t \sim iid(0, \sigma_e^2)$ and

$$\frac{\theta}{1+\theta^2} = \frac{-\sigma_\omega^2}{\sigma_v^2 + 2\sigma_\omega^2}$$

For a given $\sigma_v^2$, as $\sigma_\omega^2 \to \infty$, i.e. weather noise becomes large relative to the size of the steps in the forcing signal, we will have $\frac{\theta}{1+\theta^2} \to -\frac{1}{2}$ which implies $\theta \to -1$. The process thus approaches

$$\tilde{z}_t - \tilde{z}_{t-1} = e_t - e_{t-1}$$

in which $\tilde{z}_t$ becomes an *iid* process because the MA component has a unit root that cancels the autoregressive unit root. When $\sigma_\omega^2$ is large but finite relative to $\sigma_v^2$ the MA parameter $\theta$ will tend to be close to (but not equal to) -1 in which case, while a unit root test applied to $\tilde{z}_t$ should not reject, it is generally known in the time series econometrics literature that such 685    tests have a tendency to over-reject the unit root null.

It may be more realistic to allow for autocorrelation in the weather noise, which we do following NP02. As before $z_t$ is the sum of three components but this time we replace the iid term $\omega_t$ with $\lambda_t$ where $\lambda_t = \phi\lambda_{t-1} + \omega_t$, $|\phi| < 1$ and $cov(\omega_t, v_t) = \sigma_{\omega v}^2$. Then from equation (6) of NP02 we can write the first differences of the deviation term $\tilde{z}_t$ as


$$\Delta\tilde{z}_t = \phi\Delta\tilde{z}_{t-1} + u_t$$

where $u_t = e_t + \theta e_{t-1}$, $e_t \sim iid(0, \sigma_e^2)$, and

$$\frac{\theta}{1+\theta^2} = \frac{-\phi\sigma_v^2 - \sigma_\omega^2 - (1+\phi)\sigma_{\omega v}^2}{(1+\phi^2)\sigma_v^2 + 2\sigma_\omega^2 + 2(1+\phi)\sigma_{\omega v}^2}.$$





Again for a given $\sigma_v^2$ and $\sigma_{\omega v}^2$, as $\sigma_\omega^2 \to \infty$, $\frac{\theta}{1+\theta^2} \to -\frac{1}{2} \Longrightarrow \theta \to -1$. Consequently, the same outcome emerges when weather noise is autocorrelated, namely that as weather noise dominates there will be a tendency to reject the unit root null hypothesis even though a unit root is known to be present via the $\tau_t$ component.


Now suppose there is an ensemble of models denoted $i = 1, \dots, N$, which share common forcings $\tau_t$ but embed different weather processes $\omega_{it}$. Unit root tests applied to individual models will all exhibit the potential bias towards false rejection, and the average of such individual test scores will share the same bias. However, a unit root test applied to the ensemble average will not. To show this, assume the weather processes are autocorrelated and that all models have the same

autocorrelation parameter $\phi$. Then $\tilde{z}_{it} = \tau_t + \lambda_{it}$ where $\lambda_{it} = \phi\lambda_{it-1} + \omega_{it}$. The detrended series for the model ensemble mean is

$$\bar{\tilde{z}}_t = \frac{1}{N}\Sigma_{i=1}^N \tilde{z}_{it} = \tau_t + \bar{\lambda}_t$$

where $\bar{\lambda}_t = \frac{1}{N}\Sigma_{i=1}^N \lambda_{it} = \phi\bar{\lambda}_{t-1} + \bar{\omega}_t$ and $\bar{\omega}_t = \frac{1}{N}\Sigma_{i=1}^N \omega_{it}$. If model weather has the same variance across models and is also uncorrelated across models, then $var(\bar{\omega}_t) \equiv \sigma_{\bar{\omega}}^2 = \frac{1}{N}\sigma_\omega^2$. For the covariance term we obtain $cov(v_t, \bar{\omega}_t) \equiv \sigma_{\bar{\omega}v}^2 = cov\left(v_t, \frac{1}{N}\Sigma_{i=1}^N \omega_{it}\right) = \frac{1}{N}\Sigma_{i=1}^N cov(v_t, \omega_{it}) = \frac{1}{N}n\sigma_{\omega v}^2 = \sigma_{\omega v}^2$ which is the same as in the individual series cases. Now using the result in NP02 as before we can write $\Delta\bar{\tilde{z}}_t = \phi\Delta\bar{\tilde{z}}_{t-1} + u_t$ where $u_t = e_t + \theta e_{t-1}$ and

$$\frac{\theta}{1+\theta^2} = \frac{-\phi\sigma_v^2 - \sigma_{\bar{\omega}}^2 - (1+\phi)\sigma_{\bar{\omega}v}^2}{(1+\phi^2)\sigma_v^2 + 2\sigma_{\bar{\omega}}^2 + 2(1+\phi)\sigma_{\bar{\omega}v}^2}$$

$$= \frac{-\phi\sigma_v^2 - \frac{1}{N}\sigma_\omega^2 - (1+\phi)\sigma_{\omega v}^2}{(1+\phi^2)\sigma_v^2 + \frac{2}{N}\sigma_\omega^2 + 2(1+\phi)\sigma_{\omega v}^2}$$

The contribution of the variance of weather noise is now $\frac{1}{N}\sigma_\omega^2$ instead of $\sigma_\omega^2$ while all other terms are the same. Thus for the same magnitude of weather noise, $\theta$ will start farther away from -1 and the convergence to -1 will be slower. The tendency

of unit root tests to over-reject the unit root null is reduced because of averaging across model series.





**Data and Code Availability:** See McKitrick R (2022) Mendeley Data, V1, doi: 10.17632/sbhrkxks2r.1

**Author Contributions:** Study conception and methods: RM and TV. Development of testing theory: TV. Compilation of the tropospheric observational and model data sets: JC. Coding, testing and figure production: TV and RM. First draft: RM. All authors read, edited and approved the final manuscript.

**Competing interests:** No funding was received for this work. TV and JC declare that they have no competing interests. RM is an unpaid Senior Fellow of the Fraser Institute and an unpaid member of the Academic Advisory Council of the Global Warming Policy Foundation. Neither organization had any knowledge of, involvement with or input into this research. RM has provided paid or unpaid advisory services to private sector entities in the law, manufacturing, distilling, communications, policy analysis and technology sectors. None of these entities had any knowledge of, involvement with or input into this research.

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
