# Peer review of "Temperature Trends, Climate Attribution and the Nonstationarity Question"

_Earth System Dynamics, 2023_

## Referee Comment (RC1)

This paper "Temperature trends, climate attribution, and the nonstationarity question," returns to the question; are the time series for surface temperature trend stationary or do they contain a unit root (i.e. they are I(1))? As explained by the authors on page 3, the mean value of a trend stationary time series changes over time because the time series increases/decreases by the same quantity period-after-period. Conversely, the mean value of a time series that contains a unit root changes over time due to accumulation of random changes.

Differences between trend stationary and nonstationary processes are important for efforts to attribute climate change to human activity. According to the anthropogenic theory of climate change, observed changes in radiative forcing are driven by economic activities that emit climate change. These activities, along with the long residence time in the atmosphere, imply that the time series for radiative forcing contain a unit root (Kaufmann et al., 2013). These nonstationary changes should appear in the time series for temperature if human activity drives climate change (unforced temperature is not I(1)). Conversely, the time series for temperature will be trend stationary (with or without a break) if some 'unknown' deterministic process changes climate.

In this manuscript, the authors argue that the temperature time series are trend stationary after allowing for a single structural break in trend. They use this result to revisit conclusions about cointegration between time series for radiative forcing and surface temperature. Although interesting, it is difficult to interpret results and evaluate the reliability of this conclusion because the current manuscript contains several difficulties and ignores previous research.

One important issue is the time series used by the authors to represent the radiative forcing of aerosols in particular and radiative forcing in general. The forcings used by the authors are obtained from CMIP5 (page 15) and shown in their Figure 3. These time series are highly stylized (and linearized) and are used to simulate climate models, but they are very different from the time series used by statistical analyses of temperature. For example, the time series for the radiative forcing of aerosols in Figure 3b is nearly flat. This is very different from the forcing associated with anthropogenic sulfur emissions, which become increasingly negative from 1950 through the early 1970s, flatten out through the late 1990's, and becomes increasingly less negative thereafter (e.g. Figure 1, Kaufmann et al., 2011). These changes play a critical role in statistical explanations for historical changes in global temperature; temperature declines slightly between the end of WWII through the mid 1970's because the radiative forcing of anthropogenic sulfur emissions increases (in absolute terms) slightly faster than increase in the radiative forcing of greenhouse gases (Kaufmann et al., 2006). Global temperature increases rapidly thereafter when efforts to reduce acid deposition reduce the radiative forcing of anthropogenic sulfur emissions relative to the radiative forcing of greenhouse gases (Kaufmann et al., 2006) Finally, the hiatus in global temperature from 1998 to 2008 is associated with an (absolute) increase in the radiative forcing of anthropogenic sulfur emissions (Kaufmann et al., 2011).

Given these effects, the authors need to describe why they use the highly stylized forcings from CMIP5 as opposed to the forcing used by statistical analyses, which are the focus of the authors' efforts. These time series used in statistical analyses are readily available. For example, they can be downloaded from http://www.sterndavidi.com/datasite.html or they can request them from the authors of papers that they cite.

The results of the paper also are obfuscated by the way that the authors analyze the time series for temperature that are simulated by climate models (Tables 1 & 2). The authors test the temperature time series that are simulated by the "under anthropogenic-only forcing" scenario for a linear trend. These authors find a general increase in temperature. But these increases in temperature are not generated by a deterministic trend. They likely are generated by the "anthropogenic-only forcings" that are used to simulate the models. But there is no reason to believe that these increases in radiative forcing are trend stationary (with or without a break). Human activities that generate emissions of radiatively active gases are not trend stationary. Furthermore, long residence times in the atmosphere impart a unit root.

Indeed, finding that the temperature time series generated by the climate models are trend stationary with a break undermines their basic hypothesis. If the temperature generated by climate models are trend stationary with a break, that implies that the forcings used to simulate the model are trend stationary with a break. So, the authors should test the time series used to simulate the climate model are trend stationary with a break. Such a result would contradict previous analyses that indicate these time series are not trend stationary with a break (Kaufmann et al., 2013). Furthermore, there is no physical mechanisms that will generate time series for radiative forcing that re trend stationary with a break. Conversely, if the authors believe that there is something about the climate system that causes temperature to be trend stationary (with or without a break), they should test the temperature data that are generated by control scenarios, in which radaitive forcing does not systematically increase or decrease over time.

I also find that the authors ignore previous efforts to determine whether the temperature time series are trend stationary with a break or whether they cointegrate with radiative forcing. Beyond simple tests, Kaufmann et al (2010), which the authors cite, compare the in-sample accuracy for surface temperature generated by a trend stationary model (with a single break) against a simulation generated by combining cointegration and error correction models. Their results indicate the cointegration error correction approach generated a more accurate in-sample simulation of temperature than the trend stationary model with a single break. This seems to be a much more power approach than relying on a single test statistic. At a minimum, I ask the authors to compare the accuracy of their trend stationary model (with a single break) against that generated by a cointegration/error correction approach that are estimated using the time series for radiative forcing that are used by previous statistical models.

Page 2 "That temperatures are composed of a nonstationary forcing component and stationary weather noise which biases the unit root tests towards non-detection of the stochastic trend." This statement ignores a more sophisticated analysis by Stern and Kaufmann (2000). They use multivariate structural time series techniques to decompose Northern and Southern Hemisphere temperatures into stochastic trends and autoregressive noise processes (i.e. stationary weather noise). Their results show that there are two independent stochastic trends in the data. One is similar to the radiative forcing due to greenhouse gases and solar irradiance and a second trend, which represents the non-scalar non-stationary temperature differences between the hemispheres that reflects radiative forcing due to tropospheric sulfate aerosols. They confirm these results by analyzing temperature data generated by the Hadley Centre GCM SUL experiment. In short, the authors should explain how their analysis extends/supersedes Stern and Kaufmann (2000).

I also have concerns about the section on cointegration analysis. The authors focus on cointegration between global temperatures and global forcing but ignore analyses that look at hemispheric relations. Kaufmann and Stern (2002) estimate a cointegrating vector autoregression model (Juselius, 2006) and use the Johansen trace statistic to find that differences in hemispheric temperature are associated with differences in the hemispheric temperature effects of greenhouse gases, anthropogenic sulfur emissions, and solar irradiance. I also find the focus on cointegration between forcings and temperatures simulated by climate models misleading. Any failure to find cointegration likely represents failings of the climate models rather than any real-world decoupling between observed values for radiative forcing and observed temperature.

It is not fair to dismiss the results found by Dergiades et al., (2016) with the innuendo on page 9 "The transition from I(0) to I(1) behavior in the Wahl and Amman (2007) chart may thus be an artifact of the climate proxy choice." The authors need to demonstrate that the 'choice' of the "Wahl and Amman (2007) chart" creates this result by using a different proxy for temperature and repeating the analysis by Dergiades et al., (2016). Short of that, this sentence is speculation, which really does not belong in a scientific paper.

Page 3 "There is no established label for a trending series with an I(1) random component, so we adopt the label 'trend stationary-random component or the more compact label 'trend stationary' for such a process. This is incorrect. There is a well-recognized term for such a series, a random walk with drift (Enders, 1995).

Page 6 "What happens if different unit root tests yield conflicting results?" No need to ask this question, it has been answered by Stern and Kaufmann (2000). Their Table I reports results of the ADF, Phillips-Perron, Schmidt-Phillips, and KPSS test statistics as applied to both global surface temperature, temperature in the Northern Hemisphere, and temperature in the Southern Hemisphere.

**Literature Cited (not cited by McKitrick et al, 2023)**

Enders, W. 1995, Applied Econometric Time Series, John Wilyer & Sons, New York.

Juselius, K., 2006, The Cointegrated VAR Model, Oxford University Press, Oxford, UK.

Kaufmann, RK, H. Kauppi, ML Mann, and J.H Stock, 2011, Reconciling anthropogenic climate change with observed temperature 1998-2008, *Proceedings National Academy of Sciences* 108(29):11790-11793 doi/10.073/pnas.1102467108.

Kaufmann, R.K. and D.I. Stern. 2002 Cointegration analysis of hemispheric temperature relations. *Journal of Geophysical Research*. 107 D2 10.1029, 2000JD000174.

Stern, D.I. and R.K. Kaufmann, 2000, Is there a global warming signal in hemispheric temperature series: a structural time series approach *Climatic Change*. **47**:411-438.

---

## Author Comment (AC1)

**DRAFT Response to Review #1**

We appreciate the detailed review and constructive comments of the reviewer. Herein we provide responses point-by-point and explain how we will revise the paper to take them into account.

**Page 1:**
The first three paragraphs summarize the key issues. The last part of the first paragraph describes the difference between stationary and non-stationary trending series as follows:

> "the mean value of a trend stationary time series changes over time because the time series increases/decreases by the same quantity period-after-period. Conversely, the mean value of a time series that contains a unit root changes over time due to accumulation of random changes."

Since our choice of terminology differs from the reviewer's it is worth elaborating on this point. The above statement is only true for the unit root case if the random changes have non-zero mean because a unit root process driven by mean zero innovations has a constant expected value (zero), so there is no trend. If the innovations have non-zero mean, then the process does have a linear time trend with mean zero unit root innovations around the deterministic trend. The slope of the linear time trend is often called the 'drift' when the unit root process is the special case of a random walk (serially uncorrelated mean zero innovations).

What is important to keep in mind in the analysis of time series processes that exhibit a steady increase is that the increase cannot be explained by the stochastic fluctuations of a unit root process. Instead it is represented by the linear trend component. For cointegration analysis it is the relationship between the stochastic fluctuations among unit root processes that matters. The steady increase represented by the linear trend is a separate component the identification of which is not directly related to determining whether the stochastic components are $I(0)$ or $I(1)$ or whether there is cointegration. Unfortunately, the time series econometrics literature has applied the label 'stochastic trend' to mean zero unit root processes which suggests such a series can systematically increase or decrease. But its mean is constant at zero so its motions should not be called a "trend". Doing so can lead to spurious conclusions when using trend tests that assume the stochastic component is $I(0)$, which is where the label 'stochastic trend' originated. A unit root process only systematically increases or decreases if its mean is increasing or decreasing over time. The simplest example is a random walk with drift which is the sum of a linear trend and a mean zero unit root process with serially uncorrelated innovations. When the innovations are serially correlated, the label random walk with drift no longer applies, but there is no agreed-upon alternative. In such a case we use "trend nonstationary" because this label makes it clear that the series has a deterministic trend (the drift from the first differenced representation) and $I(1)$ fluctuations (but not necessarily random walk) around the trend. We avoid the label stochastic trend because that implies a mean zero $I(1)$ process that is not trending.

*Paragraphs 4&5: critique of our forcings data*
We use 2 forcings data sets, the CMIP5 input series (shown in Fig 3) and the model-generated temperature counterfactuals from C22. The referee asks why we didn't use the longer time series available from the David Stern website. One reason is that these data only go up to 2011. Another is

that they predate a decade's worth of research on the forcing strength of aerosol emissions, which have led to downward revisions of IPCC consensus estimates.

The referee notes that our aerosol series does not show a large magnitude of net forcing. Figure 3 shows each forcing with the same vertical axis for comparison. Apparent flatness of the aerosol forcing compared to the GHG forcing reflects the fact that the CMIP5 series has a smaller range of values than previous estimates. Since the IPCC AR4 in 2007, the scientific literature has revised the strength of aerosol cooling downwards (or, since the forcing estimate is negative, revised the central estimate upward). The IPCC AR5 WG1 noted this change in the Working Group I report of the AR5 (IPCC 2013 p. 574).

Subsequently there have been numerous papers confirming the reduced aerosol forcing strength with particular focus on aerosol-cloud interactions turning out to be weaker than previously thought and weaker than is typically represented in many climate models. Detailed reviews are provided in Lewis and Curry (2018 see p. 6055) and in Lewis (2022) SI Section 5.2.4. Recent reconstructions of pre-industrial wildfire-related aerosol emissions have likewise reduced the implied net aerosol forcing estimates (Hamilton et al. 2018, Liu et al. 2021).

Therefore it would not make sense for us to use pre-CMIP5 aerosol forcing estimates. Even the CMIP5 aerosol forcings may be too large in absolute magnitude but, as we explain, there is no single, convenient series available in the CMIP6 archive.

Other points:
- We didn't linearize the forcings. They are centered on 0 but otherwise used as-is
- We find evidence that the forcings are I(1) or I(2) which is the same as Kaufmann and Stern so we don't see any disagreement on that point
- Our cointegration analysis uses the Cummins et al (2022) model-generated counterfactual temperature simulations rather than the forcing series themselves, based on the argument in C22 that this is a valid substitution. The specific discussion point in that section requires that we use the C22 data, and once again it is a more recent data series and would be more likely to reflect the post-2010 revision in aerosol forcing estimates.

**Page 2**
*Paragraphs 1 & 2: data selection and analysis*
The reviewer appears to have misunderstood the nature of our climate model data. The surface model runs in Figure 1 and Table 1 are all-forcing simulations, not "anthropogenic only." This is clearly stated in the text p. 10 lines 297-301. The text explains that the anthropogenic-only series will be used later in the cointegration analysis. It is not used for our main results. The tropospheric model runs (Figure 2, Table 2) are likewise "all-forcing" as is explained on page 12 lines 343-344.

The reviewer raises the question of how the model simulated temperatures could appear to be trend+break with I(0) errors if the anthropogenic forcing inputs are I(1), and suggests this undermines our basic hypothesis because both series have to have the same order of integration, specifically saying "If the temperature generated by climate models are trend stationary with a break, that implies that the forcings used to simulate the model are trend stationary with a break." We disagree with this claim on the grounds elaborated in the theoretical section of our paper. Climate models combine stationary and nonstationary components. We discuss at length in Section 2.2 and Appendix B the challenge of testing a series composed of underlying elements with

different orders of integration. The property of the summed series will depend on which component is dominant. In Appendix B we show how a large amount of I(0) noise overlaid on a smooth I(1) series can cause a unit root test to be biased towards rejecting the I(1) null, and if this is the case, de-noising in the form of averaging out the I(0) noise process should move the test results towards non-rejection. Our empirical results provide evidence for this effect in the model data case, since the de-noised model data moves towards the non-rejection region, although we still reject the unit root null against a broken trend alternative. No such effect is observed with the observational data. Regarding the forcings, we agree that they are I(1) or I(2).

We are aware that our test scores on model data contradict previous findings, such as Kaufmann et al. (2013). It should be noted that we are using different models, different tests and a different time interval so obtaining the same results is not guaranteed. We agree however that we did an inadequate job of positioning our work in the context of previous findings and explaining why our results differ in some cases. We will remedy this by including a more extensive discussion of relevant prior studies, including Gay-Garcia et al. (2009), Kaufmann et al. (2010) & (2013), Mills (2010) and Estrada and Perron (2019).

*Paragraph 3: contrast with earlier literature*
The reviewer is correct that we did not adequately discuss the earlier literature on whether temperatures are trend-with-break+I(0) errors or nonstationary. Specifically the findings of Gay-Garcia et al. (2009) and the subsequent debate with Kaufmann et al. (2010) and the comment of Mills (2010) are relevant, as is the recent work of Estrada and Perron (2019). The latter, for instance, argues that both temperatures and forcings are trend-with-break+I(0), whereas most climate econometrics authors follow Kaufmann et al. (2010) in arguing that both are I(1). We find that temperatures are trend-with-break+I(0) while anthropogenic forcings are I(1) or I(2) and propose that this is the puzzle needing to be explained, if it is assumed that temperatures inherit the stationarity properties of anthropogenic forcings. Our theoretical and empirical results propose one possible explanation.

The reviewer refers to the in-sample test used in K10, in which they found that in the NH (but not in the SH) a cointegrating model based on I(1) temperatures achieves a better in-sample fit than the trend-with-break+I(0) errors. Such a test is more pertinent in forecasting applications, but that is not what we are doing. We are using the trend+break specification as an approximation to the steady increase in temperatures and forcings for the purposes of studying whether the random fluctuations are better characterized as I(0) or I(1). This would be a standard preliminary analysis before building a forecasting model that relates temperatures to forcings and would be informative as to whether or not a cointegrating model is reasonable. It's not surprising that the cointegration model used by Kaufmann et al (2010) gives a better fit than the trend+break model. The trend+break fitted model in their Figure 1 does not model any dynamics in the random component (for instance lagged temperatures) and does not include covariates that can help explain temperatures. The cointegration model has both and would be expected to give a better fit regardless of whether or not there is cointegration in the stochastic components.

*Paragraph 4: contrast with S&K 2000*
The analysis in Stern and Kaufmann (2000) applies a different technique (structural time series analysis) on a different data set and answers different questions. We don't think it is necessary, in principle, for every study on the topic to use the same methods, as long as the methods used in a study are appropriate for the questions being asked and the data being used, which is the case in

our study. However the substantive question raised by S&K2000, namely whether the NH and SH behave differently and the absence of a property (e.g. nonstationarity) at the global level may mask its presence in the hemispheres in a form that cancels out in the aggregate, is valid. We will examine this question by re-doing our analysis on the NH alone to see if different results emerge. We are in the process of assembling this data set.

**Page 3**

*Paragraph 1: use of C22 data*
Here again the main question being posed can be addressed by redoing our analysis on NH data, which we will do.

The reviewer critiques our use of climate model-generated data for the cointegration analysis in Section 3.5. But these are the data used in Cummins et al. 2022 and it is necessary for us to use them. It is valid to point out that this method presupposes the validity of climate models, but this is a weakness of many attribution methods in climatology. We find evidence of cointegration of the model series with each other but not with temperatures. This finding is useful for reconciling our results with the stationarity of the error terms in the C22 regression model, which they interpreted to mean successful signal detection but without having established that the observed temperatures themselves were I(1). The reviewer makes a rather strong assertion in claiming that any failure to find cointegration must mean that the models that generated the data are invalid, since the data must be cointegrated. This amounts to assuming the conclusion. Furthermore, if the models are indeed invalid, the problem may be that they overstate the coupling between forcings and temperature, in which case stationarity of the observed temperatures would be expected.

*Paragraph 2: critique of Dergiades result*
This is a fair comment. The paragraph will be removed. However we need to address the Dergiades result and we will do so in a more formal way. The Mann (equivalently Wahl and Ammann) reconstruction is available back to 1000 AD. We can replicate the Dergiades result on the post-1700 segment showing that the unit root test wanders into the non-rejection region. However it spends long intervals in the non-rejection region in the pre-1600 interval as well, even though the underlying hypothesis (that non-rejection is due to the emerging dominance of anthropogenic forcings) rules this out. This raises the question of whether the proxy reconstruction is suitable for the purpose, and here it is valid to point out that the reconstruction is a splice of many segments composed of differing proxy rosters, and the discontinuities may cause problems for the types of tests being used, which is a point raised by Pretis and Hendry (2013) in a similar context.

*Paragraph 3: terminology*
The reviewer is correct that a random walk with drift is the conventional label for a time series process that is the sum of a linear deterministic trend and a random walk stochastic process. But, a random walk is a special case of a unit root process in which the innovations are serially uncorrelated. This is why we prefer the label trend-nonstationary because it does not imply the random walk special case. The 'trend' part of the label refers to the deterministic trend and the 'nonstationary' part of the label refers to the mean zero unit root stochastic component which may be serially correlated.

*Paragraph 4: no need to ask about conflicting results*
It's not clear what is the objection to asking the question, since obtaining conflicting test results is very common. We ask the question to motivate the discussion of the required steps in constructing a valid unit root test and discriminating among conflicting inferences especially with respect to choice of lag length for the ADF regressions.

Sources:

Cummins, D., D.B. Stephenson and P.A. Stott. (2022). "Could detection and attribution of climate change trends be spurious regression?" *Climate Dynamics* March 2022 https://doi.org/10.1007/s00382-022-06242-z

Estrada, F and P Perron (2019) Breaks, Trends and the Attribution of Climate Change: A Time-Series Analysis. *Economia* 42(1) 1—31 https://doi.org/10.18800/economia.201901.001

Gay-Garcia C., Estrada F., and Sanchez A. (2009). Global and hemispheric temperature revisited. Climatic Change 94(3-4), 333-349

Hamilton, DS, S. Hantson, CE Scott et al. (2018) Reassessment of pre-industrial fire emissions strongly affects anthropogenic aerosol forcing. *Nature Communications* 2018 9:3182, DOI: 10.1038/s41467-018-05592-9 |

IPCC Fourth Assessment Report (2007), Working Group I

IPCC Fifth Assessment Report (2013), Working Group II

Kaufmann, R.K., Kauppi, H. and J. Stock (2010). Does temperature contain a stochastic trend? Evaluating conflicting statistical results. *Climatic Change* 118, 729–743. https://doi.org/10.1007/s10584-012-0683-2

Kaufmann, R.K., Kauppi, H., Mann, M.L. et al. (2013). Does temperature contain a stochastic trend: linking statistical results to physical mechanisms. *Climatic Change* 118, 729–743. https://doi.org/10.1007/s10584-012-0683-2

Lewis, N., and J. Curry, 2018. The Impact of Recent Forcing and Ocean Heat Uptake Data on Estimates of Climate Sensitivity. Journal of Climate https://doi.org/10.1175/JCLI-D-17-0667.1

Lewis N (2022) Objectively combining climate sensitivity evidence. *Climate Dynamics* https:// doi. org/ 10. 1007/s00382- 022- 06468-x

Liu, Pengfei et al. (2021) Improved estimates of preindustrial biomass burning reduce the magnitude of aerosol climate forcing in the Southern Hemisphere. *Science Advances* 7(22) DOI: 10.1126/sciadv.abc1379 https://www.science.org/doi/10.1126/sciadv.abc1379

Mills, T. (2010a). 'Skinning a cat': alternative models of representing temperature trends. An editorial comment. Climatic Change 101(3), 415-426.

Pretis, F. and D.F. Hendry (2013). Comment on "Polynomial cointegration tests of anthropogenic impact on global warming" by Beenstock et al. (2012) – some hazards in econometric modelling of climate change. *Earth System Dynamics* 4:375—84 https://doi.org/10.5194/esd-4-375-2013.

---

## Author Comment (AC2)

Response to Referee #2

*Paragraph 2:*
The reviewer suggests that we explore the shape of the distribution of our temperature data. The best way for us to do this is by drawing histograms using the detrended series, so we will create a supplement for this purpose. However, it is not necessary to establish Normality of all our temperature or residual series since the estimation and testing procedures we apply rely on asymptotic statistical theory, and in particular central limit theorems, which provide asymptotically valid critical values even if the underlying data series are not Gaussian.

The referee asks: "Which type of link is possible to derive between this model and the developing of the data in the sequence? This information is fundamental to understand something about the dynamics governing the investigate system and to intercept critical point (tipping points?) in the sequence." We find the question somewhat unclear, but we assume the referee is asking whether the trend model we employ is consistent with the underlying characteristics of the data itself. This is, indeed, the central question of the literature we are engaging with. As we discuss in the introduction, the IPCC routinely employs a trend model that the climate econometrics literature says is incompatible with the dynamics governing the temperature system, and within the climate econometrics literature there are conflicting claims about what model would be valid. We argue that a trend stationary representation similar to the one used by the IPCC is valid, but it calls into question the validity of the cointegration approach used in econometric-based attribution studies.

The referee points to a few other topics like tipping points and fractal processes. While interesting they are not directly related to our paper or the literature we are discussing so we are not able to review those topics. The types of trend breaks we are modeling may have some relation to bifurcations in dynamical systems but that's well beyond our scope.

*Paragraph 3:*
We are puzzled why the referee says autocorrelation was not sufficiently investigated. We discuss it in Sections 2.1, 3.2 and 3.3, and we present estimates in Figures 4—6. At every point in the analysis where autocorrelation matters we discuss it and take it into account, especially with regard to lag length selection in the unit root testing procedure. Although we did not report all the autocorrelation coefficients we computed (they number in the hundreds) we assure the referee that our treatment of this topic is exhaustive.